# Maximizing Feature Distribution Variance for Robust Neural Networks

## ABSTRACT

The security of Deep Neural Networks (DNNs) has proven to be critical for their applicabilities in real-world scenarios. However, DNNs are well-known to be vulnerable against adversarial attacks, such as adding artificially designed imperceptible magnitude perturbation to the benign input. Therefore, adversarial robustness is essential for DNNs to defend against malicious attacks. Stochastic Neural Networks (SNNs) have recently shown effective performance on enhancing adversarial robustness by injecting uncertainty into models. Nevertheless, existing SNNs are still limited for adversarial defense, as their insufficient representation capability from the fixed uncertainty. In this paper, to elevate feature representation capability of SNNs, we propose a novel yet practical stochastic neural network that maximizes feature distribution variance (MFDV-SNN). In addition, we provide theoretical insights to support the adversarial resistance of MFDV, which primarily derived from the stochastic noise we injected into DNNs. Our research demonstrates that by gradually increasing the level of stochastic noise in a DNN, the model naturally becomes more resistant to input perturbations. Since adversarial training is not required, MFDV-SNN does not compromise clean data accuracy and saves up to 7.5 times computation time. Extensive experiments on various attacks demonstrate that MFDV-SNN improves adversarial robustness significantly compared to other methods.

## CCS CONCEPTS

• **Security and privacy** → *Human and societal aspects of security and privacy*; • **Computing methodologies** → *Artificial intelligence*.

## KEYWORDS

Model Robustness, Model Uncertainty, Adversarial Defense

## 1 INTRODUCTION

Despite the promising performance of deep learning models, recent studies have shown that Deep Neural Networks (DNNs) are vulnerable to adversarial attacks [5, 10]. The typical attacks, such as adding minor, carefully crafted perturbations to model inputs, can potentially cause misguided and incorrect decisions for DNNs, which is imperceptible to human recognitions. This poses a serious security threat to the practical application of deep models. To

**Unpublished working draft. Not for distribution.**

Permission to make digital or hard copies of all or part of this work for personal or classroom use is granted without fee provided that copies are not made or distributed for profit or commercial advantage and that copies bear this notice and the full citation on the first page. Copyrights for components of this work owned by others than the author(s) must be honored. Abstracting with credit is permitted. To copy otherwise, or republish, to post on servers or to redistribute to lists, requires prior specific permission and/or a fee. Request permissions from permissions@acm.org.

*ACM MM, 2024, Melbourne, Australia*

© 2024 Copyright held by the owner/author(s). Publication rights licensed to ACM.
ACM ISBN 978-x-xxxx-xxxx-x/YY/MM
https://doi.org/10.1145/nnnnnnn.nnnnnnn

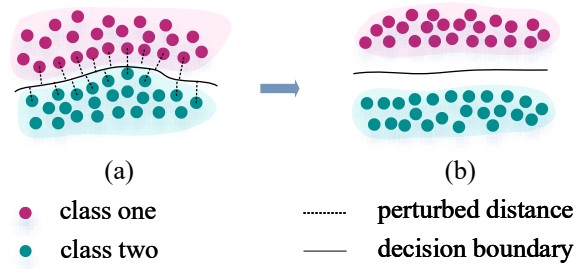

**class one**

**class two**

**perturbed distance**

**decision boundary**

**Figure 1: (a) represents a standard trained neural network, while (b) depicts the same network, but with increased feature perturbations introduced during training.**

tackle the challenge, researchers have intensified efforts to develop defense methods to enhance the robustness of these models against adversarial attacks.

Recently, Stochastic Neural Networks (SNNs) have demonstrated significant potential for model robustness by introducing uncertainty into model feature activations, such as in Variational Autoencoders [18], or into model weights, as exemplified by Bayesian Neural Networks[13]. However, *current implementations of SNNs predominantly utilize Gaussian distributions with constant variance to model this uncertainty* and *how the magnitude of the injected noise affects adversarial robustness tends to be ignored.*

However, we assume that this limitation restricts the potential to achieve higher levels of adversarial robustness. The intuitive motivation behind our approach is illustrated in Figure 1, which conceptually represents how our noise injection method is designed to improve the model resistance against adversarial attacks.

It is crucial to emphasize that the essence of adversarial robustness lies in minimizing the chances of a model misclassifying an input, especially when faced with adversarial perturbations. Figure 1(a) illustrates the decision boundaries of a standard trained neural network. Despite the model's high accuracy in standard classification scenarios, its decision boundaries are shown to be vulnerable to adversarial perturbations. In such configurations, samples near the decision boundary can easily cross it, thereby becoming adversarial examples that are misclassified. In contrast, Figure 1(b) depicts the training process under our proposed stochastic feature perturbation scheme. During training, we persistently perturb the feature embeddings and progressively increase the perturbation intensity. Initially, this leads to classification errors.

To correctly classify these perturbed instances, the model must adjust its internal parameters. Hence, we promote it to establish decision boundaries that are more robust to input variations. To be specific, at the beginning of the training, the model learns the primary patterns from the data. As training progresses and the

noise intensity increases, the model becomes increasingly resilient to these perturbations. This ensures that foundational learning isn't overshadowed by the noise. In this way, the model leverages noise as a regularizer. As a result, it not only enhances both the model's robustness and generalization capabilities but also maintaining a stable and controlled training process. Benefited from these, the new boundaries exhibit increased resilience to adversarial perturbations. We have named this model the Maximizing Feature Distribution Variance Stochastic Neural Network (MFDV-SNN).

In this work, our primary contribution is three-fold:

- To the best of our knowledge, we are the first to explore how the magnitude of the injected noise affects adversarial robustness in SNNs.
- We theoretically show that by gradually increasing the level of stochastic noise in a DNN, the model naturally becomes more resistant to input perturbations.
- The proposed method does not require adversarial training and does not sacrifice clean data accuracy. Compared to adversarial training, the computation cost saves up to average 7.5 times.

We conduct experiments on various white- and black-box attacks to demonstrate the effectiveness of the proposed MFDV-SNN compared to stochastic or non-stochastic defenses.

## 2 RELATED WORK

### 2.1 Adversarial Attack

Researchers divide adversarial attacks into white-box and black-box attacks according to whether they could obtain the gradient information.

**White-box attack:** White-box attacks mean that the attacker knows the model gradient information. A simple yet effective white-box attack method is called Fast Gradient Sign Method (FGSM) [10], which adds a small perturbation in the direction of the sign of the gradient updates. It can be formulated as

$$\vec{x}' = \vec{x} + \epsilon \cdot \text{sign} \left( \nabla_{\vec{x}} \mathcal{L}(h(\vec{x}), y) \right) \quad (1)$$

where $\vec{x}$ denotes the input image, $\epsilon$ denotes the perturbation strength, $\mathcal{L}$ denotes the loss function and $h(\cdot)$ denotes the target model. Kurakin further updated the one-step attack FGSM to a multi-step attack named Basic Iterative Method (BIM) [19]. Compared to FGSM, BIM uses a smaller step size to explore the possible adversarial direction. Madry further updates BIM by randomly initializing the input point, which is one of the strongest first-order attacks named Projected Gradient Descent (PGD) [23]. It can be formulated as

$$\vec{x}^{t+1} = \Pi_{\vec{x}+s} \left( \vec{x}^t + \alpha \cdot sign \left( \nabla_{\vec{x}} \mathcal{L} \left( h(\vec{x}^t, y) \right) \right) \quad (2)$$

where $\Pi_{\vec{x}+s}$ is the projection operation that force the adversarial example in the $\ell_p$ ball $s$ around $\vec{x}$, and $\alpha$ is the step size. Another strong first-order attack algorithm is called C&W attack [5] which finds adversarial examples by solving the following optimization function formulated as

$$\min \left[ \|\delta\|_p + c \cdot h(\vec{x} + \delta) \right] \text{ s.t. } \vec{x} + \delta \in [0, 1]^n \quad (3)$$

where $p$ is the norm distance, commonly choosing from $\{0, 2, \infty\}$.

**Black-box attack:** Unlike white-box attacks, black-box attackers can only access the model through queries. There are mainly two ways to fool a model. One is to train a substitute of the model, in which attackers query from the target model and generate a synthesized dataset with input and the corresponding output. Due to the transferability of adversarial examples, attackers can attack alternative and target models. The limitation of this method is that it cannot execute multiple queries in reality. The other is to estimate the gradients via multiple queries to the targeted model [27]. Among them, zero-order optimization [6] algorithm aims to estimate the gradients of the target model directly.

In this study, we use well-known white and black box attacks to evaluate our method, including FGSM, $PGD_{10}$, C&W, $PGD_{100}$, n-pixel attacks, and Auto attack [7].

### 2.2 Stochastic Defense

To improve the model's robustness against unseen attacks. Stochastic defenses are proposed which mainly introduce stochastic noises into model weights/activations to simulate possible model parameters and the related probability distribution. Typically stochastic neural networks, such as Bayesian neural networks, introduce randomness into model weights and transform the parameter point estimates into distribution estimates. Unlike introducing randomness to model weights, methods such as Variational auto-encoders (VAE) [18] improve model robustness by introducing randomness to the model feature activations [32]. Our proposed method belongs to the latter, and similar defense methods are as follows.

RSE [21] uses ensemble tricks to improve model robustness. Specifically, they inject standard Gaussian noise into multi-layers during training and then perform multiple forward passes to test it. It means they only need training once and can be considered an ensemble model. Adv-BNN [22] improves the robustness of the model by introducing a standard isotropic Gaussian noise prior to the model weights and further using adversarial training to find the best model distribution. However, the above methods are all use fixed noise parameters, and hence the uncertainty of the model is fixed. Parameter noise injection (PNI) [12] further propose learning a sensitive parameter to control the variance by adding trainable Gaussian noise to each layer of features or weights of the model. Moreover, L2P [16] updates PNI by introducing a perturbation injection module and alternating training the perturbation injection module and the neural network module, which they called "alternating back-propagation," to improve the robustness of the model.

Our method is based on a stochastic neural network framework and aims to explore the effect of noise injection on adversarial robustness, and the results of the study can be extended to existing stochastic neural network methods.

## 3 METHODOLOGY

In this section, we detail the implementation of the proposed MFDV-SNN. The critical point is shown in Figure 2.

### 3.1 Implementation of Stochastic Layer

Figure 2 illustrates the process of constructing a stochastic layer. The data passes through a neural network to get $h_l$, $h_l \in \mathbb{R}^{N \times M}$. We build the Gaussian distribution by keeping the original feature $h_{l+1}$ plus a zero-mean matrix Gaussian distribution. In practice, we use non-informative prior to initializing the Gaussian variance by

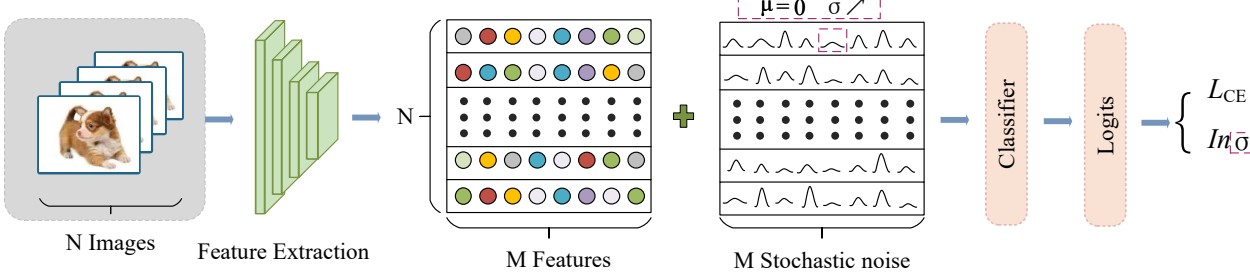

**Figure 2: Illustration of the proposed MFDV-SNN model. Input data is processed through a feature extraction layer to capture essential data features. These features are then merged with stochastic noise, characterized by a mean of zero and a progressively increasing variance during training. Samples created from this resulting distribution are directed through a classifier layer, yielding the final logit scores.**

sampling the same dimension with the feature dimension from the uniform distribution. Then, we establish a Gaussian distribution $z$ and sample from it. Finally, we take the plus of the sample and the original hidden representation $h_{l+1}$ to get the logit of the network.

## 3.2 Analysis of Noise Injection Regularization

In this section, we show that introducing additional stochastic noise into a deep neural network (DNN) can naturally create an effective regularization mechanism to counteract input perturbations.

Consider a standard DNN used for supervised learning. For a single sample, the loss function $\mathcal{L} : \mathbb{R}^{d_{in}} \to \mathbb{R}$ is defined. The loss depends on model parameters $\theta = W^{(\ell)}, x^{(\ell)}, b^{(\ell)} | \ell = 0, ..., N_L - 1$, where $N_L$ is the number of layers, with weights, features, and biases of a given layer being $W^{(\ell)} \in \mathbb{R}^{d_\ell \times d_{\ell+1}}$, $x^{(\ell)} \in \mathbb{R}^{d_\ell}$, $b^{(\ell)} \in \mathbb{R}^{d_{\ell+1}}$, respectively. In each optimizer iteration, a mini-batch $\mathcal{B}$ includes a set of labeled examples, $(x_i, y_i) i = 1^{|\mathcal{B}|} \in \mathbb{R}^{din} \times \mathbb{R}^{d_{label}}$.

Adding stochastic noise (SN) in a given layer $\ell_{SN}$ corresponds to a random scalar input, $\epsilon \in \mathbb{R}$. The batch average loss function with SN can be written as a series expansion of the noise shift parameter $\epsilon W_{SN}$.

Introducing noise injection in a specific layer $\ell_{SN}$ transforms the activation to $z^{(\ell_{SN})} + W_{SN}\epsilon$. The single sample loss function for transformed activation is expressed as

$$\mathcal{L}(\theta, W_{SN}; x, \epsilon, y) = \mathcal{L}\left(\theta; z^{(\ell_{SN})} + W_{SN}\epsilon, y\right). \quad (4)$$

Using the definition of the translation operator $f(x + a) = e^{a\nabla} f(x)$, we can explicitly compute the batch average loss function

$$
\begin{aligned}
L(\theta, W_{SN}) &= \frac{1}{|\mathcal{B}|} \sum_{\{x, \epsilon, y\} \in \mathcal{B}} \mathcal{L}(\theta, W_{SN}; x, \epsilon, y) \\
&= \frac{1}{|\mathcal{B}|} \sum_{\{x, \epsilon, y\} \in \mathcal{B}} e^{\epsilon W_{SN}^T \nabla_{z^{(\ell_{SN})}}} \mathcal{L}(\theta; x, y) \\
&= L(\theta) + \frac{1}{|\mathcal{B}|} \sum_{\{x, \epsilon, y\} \in \mathcal{B}} \sum_{k=1}^{\infty} \frac{1}{k!} \left(\epsilon W_{SN}^T \cdot \nabla_{z^{(\ell_{SN})}}\right)^k \mathcal{L}(\theta; x, \epsilon, y).
\end{aligned}
$$
$$(5)$$

Here, $L(\theta)$ is the loss function without any stochastic noise (SN), and $\mathcal{R}_k$ represents the batch average derivative of the loss function with respect to activations prior to the noise injection layer.

$$\mathcal{R}_k(\theta, W_{SN}) \equiv \frac{1}{|\mathcal{B}|} \sum_{\{x, \epsilon, y\} \in \mathcal{B}} \frac{\left(\epsilon W_{SN}^T \cdot \nabla_{z^{(\ell_{SN})}}\right)^k}{k!} \mathcal{L}(\theta; x, y) \quad (6)$$

These functions are the products of moments of the injected noise, values of $W_{SN}$, and derivatives of the loss function's activation without noise injection.

Considering noise sampled from a zero-mean distribution, under this assumption, the first two terms simplify to the following forms

$$\mathcal{R}_1 = W_{SN}^T \cdot \langle \epsilon g_{\ell_{SN}} \rangle, \quad \mathcal{R}_2 = \frac{1}{2} W_{SN}^T \langle \epsilon^2 \mathcal{H}_{\ell_{SN}} \rangle W_{SN} \quad (7)$$

Here, batch average is denoted by $\langle \cdots \rangle$, and

$$g_{\ell_{SN}} = \nabla_{z^{(\ell_{SN})}} \mathcal{L}(\theta, x, y), \quad \mathcal{H}_{\ell_{SN}} = \nabla_{z^{(\ell_{SN})}} \nabla_{z^{(\ell_{SN})}}^T \mathcal{L}(\theta, x, y), \quad (8)$$

are the network-dependent local gradients and local Hessian, respectively.

Under this condition, $\mathcal{R}_1$ and $\mathcal{R}_2$ act as follows: $\mathcal{R}1$ induces constrained random walks on the norms of noise injection weights and on the data weights in layers $\ell > \ell_{SN}$, with step sizes varying according to local gradients during training. On the other hand, $\mathcal{R}_2$ can be understood as a direct regularization term on the local Hessian, striving to reduce its eigenvalues, thereby reducing local curvature between layers. These results imply that when noise is large, regularization through $\mathcal{R}_2$ becomes the dominant effect, which is also a benefit of increasing stochastic noise.

Next, we show why pushing the local Hessian towards smaller eigenvalues is beneficial for reducing sensitivity to noise perturbations.

Consider a non-random network but with noisy input, represented as $x \to x + \delta$, where $\delta$ is a random vector. Similar to the above, we can transform the pre-activation as $z^{(0)} \to z^{(0)} + W^{(0)}\delta$.

$$L(\theta)|_{x \to x+\delta} = \frac{1}{|\mathcal{B}|} \sum_{\{x, y\} \in \mathcal{B}} e^{\delta^T W^{(0)} \cdot \nabla_{z^{(0)}}} \mathcal{L}(\theta; x, y) \quad (9)$$

Table 1: Comparison of state-of-the-art SNNs for FGSM and PGD attacks on CIFAR-10 dataset with various network sizes and capacities. The proposed MFDV-SNN is compared with Parametric Noise Injection (PNI) method [12] , Adv-BNN [22] and Learn2Perturb (L2P) [16]. Results show that the proposed MFDV-SNN is effective in training robust model and achieves state-of-the-art results.

| Model | #Parameter | PNI | | | Adv-BNN | | | L2P | | | MFDV-SNN (Ours) | | |
|---|---|---|---|---|---|---|---|---|---|---|---|---|---|
| | | Clean | PGD | FGSM | Clean | PGD | FGSM | Clean | PGD | FGSM | Clean | PGD | FGSM |
| ResNet-20 | 269722 | 84.9 | 45.9 | 54.5 | 65.8 | 45.0 | 51.6 | 83.6 | 51.1 | 58.4 | 90.1 | 57.1 | 62.4 |
| ResNet-32 | 464,154 | 85.9 | 43.5 | 51.5 | 63.0 | 54.6 | 50.3 | 84.2 | 54.6 | 59.9 | 90.7 | 57.4 | 62.7 |
| ResNet-44 | 658,586 | 84.7 | 48.5 | 55.8 | 76.9 | 54.6 | 58.6 | 85.6 | 54.6 | 61.3 | 91.2 | 64.8 | 66.6 |
| ResNet-56 | 853,018 | 86.8 | 46.3 | 53.9 | 77.2 | 54.6 | 57.9 | 84.8 | 54.6 | 61.5 | 91.4 | 71.6 | 75.0 |
| ResNet-20[1.5×] | 605,026 | 86.0 | 46.7 | 54.5 | 65.6 | 28.1 | 36.1 | 85.4 | 53.3 | 61.1 | 92.1 | 66.9 | 70.1 |
| ResNet-20[2×] | 1,073,962 | 86.2 | 46.1 | 54.6 | 79.0 | 53.5 | 58.3 | 85.9 | 54.3 | 61.6 | 92.0 | 73.6 | 77.7 |
| ResNet-20[4×] | 4,286,026 | 87.7 | 49.1 | 57.0 | 82.3 | 52.6 | 59.0 | 86.1 | 55.8 | 61.3 | 93.4 | 77.0 | 83.7 |
| ResNet-18 | 11,173,962 | 87.2 | 49.4 | 58.1 | 82.2 | 53.6 | 60.0 | 85.3 | 56.1 | 62.4 | 93.7 | 79.6 | 85.7 |

Assuming the vector $\delta$ is sampled from the distribution $\mathcal{N}\left(0, \sigma_\delta^2\right)$, similar to the aforementioned, the first two terms of the Taylor expansion can be expressed as

$$\mathcal{R}_1 = \left\langle \delta^T W^{(0)} \cdot g_0 \right\rangle, \quad \mathcal{R}_2 = \frac{1}{2}\sigma_\delta^2 \operatorname{Tr}\left(\left(W^{(0)}\right)^T \langle \mathcal{H}_0 \rangle W^{(0)}\right) \quad (10)$$

It can be seen that if the noise injection reduces $\mathcal{H}_0$, it will also correspondingly decrease the sensitivity to data perturbations.

Therefore, in the mechanism of stochastic noise injection, when $\sigma$ is initially small, $\mathcal{R}_1 >> \mathcal{R}_2$, and at this point, noise injection mainly follows a random walk with small steps without significantly affecting the network's behavior. As the injected noise $\sigma$ gradually increases, $\mathcal{R}_2$ begins to surpass $\mathcal{R}_1$ and dominates in the training process. During this phase, the model's robustness against adversarial defenses to input perturbations significantly increases.

### 3.3 Loss Function

As mentioned above, we need to improve the model uncertainty during model training. In practice, we do not assign parameter variances directly because we do not know the exact uncertainty required by the model, which is closely related to the model architecture and dataset. Instead, we initialize the Gaussian variance with a non-informative standard uniform prior. When the network trains, the variance gradually increases in small steps. Also, the unbounded variance does not collapse because the gradient is $-\frac{1}{\sigma}$, and once the variance is too large, it will not back-propagate the gradient information.

Thus, the loss function can be formulated as

$$\mathcal{L} = \mathcal{L}_C - \lambda_1 \sum_{i=1}^{D} \ln\left(\vec{\sigma}_i\right) + \lambda_2 \vec{w}^T \vec{w}, \quad (11)$$

where $\mathcal{L}_C$ is the cross-entropy loss, and D denotes the feature dimension of the penultimate layer. Furtherly, we adopt the $\ln\left(\vec{\sigma}_i\right)$ operation in practice, which will facilitates the computation of gradient derivatives, and slows down the numerical change of variance. The final $L_2$ regularization term penalizes over large weights. In addition, $\lambda_1$ and $\lambda_2$ control the power of variance and weight penalty, respectively. The algorithm is shown in supplementary material.

## 4 EXPERIMENTS

### 4.1 Datasets & Adversarial Attacks

Six datasets are used in our experiments: MNIST, SVHN, CIFAR-10, and CIFAR-100, Tiny-ImageNet and Imagenette.

The MNIST dataset consists of 60K training data and 10K testing data of digit images. The SVHN dataset consists of 73K training data and 26K testing data size 32x32x3 with ten classes. The CIFAR-10 dataset consists of 50K training data and 10K testing data of size 32x32x3 with ten classes. For the CIFAR-100 dataset, the size of training data and testing data is the same as CIFAR-10, but with one-hundred classes. Tiny-ImageNet contains 100K images of 200 classes (500 for each class) downsized to 64×64 colored images. Each class has 500 training images, 50 validation images, and 50 test images. Imagenette is a subset of ImageNet with 10 classes and full-resolution images. For adversarial attack methods, we adopt various well-known white-box and black-box methods to evaluate the proposed method. Specifically, we have used FGSM attack, $PGD_{10}$ attack, C&W attack, $PGD_{100}$ attack, n-Pixel attacks, and Auto-Attack [7].

### 4.2 Experimental Setting

In this subsection, we detail the experimental setting, specifically, the attacks setting partically following the literature [12, 16]. the attacks strength $\epsilon$ of FGSM and PGD are set as 8/255. For $PGD_{10}$ attack, the steps $k$ is set as 10 and the $\alpha$ is set as $\epsilon/10$. For C&W attack, the learning rate $\alpha$ is set as $5 \cdot 10^{-4}$, the number of iterations $k = 1000$, initial constant $c = 10^{-3}$, and maximum binary steps $b_{max} = 9$. For the n-Pixel attack, the population size $N = 400$ and maximum number $k_{max} = 75$ same as [16], and we conduct a stronger 5-pixel attack which is not implemented in their setting, to validate the strong adversarial robustness of proposed method.

Besides, we choose $PGD_{100}$ and Auto-Attack as stronger white- and black-box attack methods which are not implemented in their experiments and apply different attack strength. Note that Auto-Attack contains both white- and black-box attacks. We evaluate them individually. For $PGD_{100}$ attack, we set $k = 100$ and the $\alpha$ is set as $\epsilon/100$. For Auto-Attack, we refer to the implementation from [7]. All experiments are performed on the Pytorch platform and partial

**Table 2: Comparison of state-of-the-art SNNs for white-box C&W attack and black-box n-Pixel attack on CIFAR-10 with a ResNet-18 backbone.**

|  | Strength | No Def | AT | Adv-BNN | PNI | L2P | **MFDV** |
|---|---|---|---|---|---|---|---|
|  | Clean | 92.9 | 85.5 | 82.2 | 87.2 | 85.3 | **93.7** |
| C&W | k=0 | 0.0 | 0.0 | 78.9 | 66.9 | 83.6 | **88.8** |
| | k=0.1 | 0.0 | 0.0 | 78.1 | 66.1 | 84.0 | **87.9** |
| | k=1 | 0.0 | 0.0 | 65.1 | 34.0 | 76.4 | **87.2** |
| | k=2 | 0.0 | 0.0 | 49.1 | 16.0 | 66.5 | **86.6** |
| | k=5 | 0.0 | 0.0 | 16.0 | 0.1 | 34.8 | **83.5** |
| n-Pixel | 1 pixel | 23.4 | 56.1 | 68.6 | 50.9 | 64.5 | **85.4** |
| | 2 pixels | 3.2 | 33.2 | 64.6 | 39.0 | 60.1 | **80.4** |
| | 3 pixels | 1.0 | 24.0 | 59.7 | 35.4 | 53.9 | **76.0** |
| | 5 pixels | - | - | - | - | - | **68.0** |

**Table 3: Comparison of state-of-the-art SNNs for FGSM and PGD attacks on CIFAR-10 and CIFAR-100 datasets with a ResNet-18 backbone.**

| Method | CIFAR-10 | | | CIFAR-100 | | |
|---|---|---|---|---|---|---|
| | Clean | FGSM | PGD | Clean | FGSM | PGD |
| AT [23] | 85.5 | 52.5 | 43.9 | 58.0 | 25.0 | 20.5 |
| Adv-BNN [22] | 82.2 | 60.0 | 53.6 | 50.0 | 30.0 | 27.0 |
| PNI [12] | 87.2 | 58.1 | 49.4 | 61.0 | 27.0 | 22.0 |
| L2P [16] | 85.3 | 62.4 | 56.1 | 58.0 | 30.0 | 26.0 |
| SE-SNN [33] | 92.3 | 74.3 | - | - | - | - |
| IAAT [31] | - | - | - | 63.9 | - | 18.5 |
| **MFDV-SNN (Ours)** | **93.7** | **85.7** | **79.6** | **69.4** | **47.1** | **37.3** |

attack algorithms follow foolbox[1], a public attack library. All the experiments are conducted on NVIDIA RTX 3090.

## 4.3 Comparison with Stochastic Defenses

**Baselines:** The stochastic defense baselines are as follows. Note that all these stochastic defenses need adversarial training. *No defense* [11]: models without stochastic injection served as baseline. *AT* [23]: models with adversarial training. *Adv-BNN* [22]: A combination of Bayesian neural network and adversarial training. *PNI* [12]: Injecting Gaussian noise to multilayers. *L2P* [16]: Updating PNI by learning a perturbation injection module and alternating training the noise and network module. *SE-SNN* [33]: Introducing a margin entropy loss. Moreover, there are partial comparisons against *IAAT* [31].

**White-box Attacks Results:** The white-box attacks include FGSM attack, $PGD_{10}$ attack, C&W attack, $PGD_{100}$ attack, and Auto-Attack. The results are reported in Table 1, Table 2 (Upper), Table 3, and Table 4.

**Table 4: Evaluating proposed MFDV-SNN against stronger $PGD_{100}$ and Auto-Attack, with a ResNet-18 backbone on CIFAR-10 dataset.**

| | $\epsilon/255$ | Clean | $2^0$ | $2^1$ | $2^2$ | $2^3$ | $2^4$ | $2^5$ | $2^6$ | $2^7$ |
|---|---|---|---|---|---|---|---|---|---|---|
| $PGD_{100}$ | No Defense | 93.0 | 42.7 | 12.2 | 3.2 | 1.5 | 0 | 0 | 0 | 0 |
| | **MFDV-SNN** | **93.6** | **85.7** | **85.1** | **83.4** | **79.2** | **63.4** | **26.4** | **9.1** | **2.4** |
| AA (Black) | No Defense | 93.0 | 59.9 | 24.6 | 2.8 | 0.3 | 0 | 0 | 0 | 0 |
| | **MFDV-SNN** | **93.2** | **92.7** | **92.7** | **91.8** | **83.5** | **55.2** | **18.6** | **9.4** | **6.2** |
| AA (White) | No Defense | 91.1 | 72.6 | 47.0 | 12.0 | 1.8 | 0 | 0 | 0 | 0 |
| | **MFDV-SNN** | **93.1** | **82.4** | **79.8** | **74.7** | **60.0** | **28.0** | **8.5** | **2.8** | **6.2** |

For Table 1, we compare the proposed method with PNI, Adv-BNN, and L2P on various network sizes and capacities. The proposed MFDV-SNN consistently achieves state-of-the-art results on clean accuracy and robust accuracy.

For Table 2 (Upper) C&W attack, we compare the common confidence level setting $k$ from [0,0.1,1,2,5]. The results show that the proposed MFDV-SNN significantly exceeds the stochastic defense methods even when the confidence level $k$ is 5. Specifically, compared to existed state-of-the-art defense PNI [12] and L2P [16], we have about 6.2%, 4.6%, 14.1%,30.2% and 140% improvement, respectively.

For Table 3, we compare the state-of-the-art SNNs for FGSM and $PGD_{10}$ attacks on CIFAR-10 and CIFAR-100 datasets with a ResNet-18 backbone. Specifically, for the CIFAR-10 dataset, we have about 1.5%, 15.3%, and 41.9% improvement against prior state-of-the-art stochastic defenses SE-SNN [33] and L2P [12]. For the CIFAR-100 dataset, which has richer classes than CIFAR-10, the proposed MFDV-SNN also achieves state-of-the-art results. Compared to existed defense methods IAAT [31], L2P [12] and Adv-BNN [22], we have about 8.6%, 57.0%, and 38.1% improvement, respectively.

For Table 4, we evaluate the proposed MFDV-SNN on stronger attacks, $PGD_{100}$ attack, and Auto-Attack (white-box). We divide Auto-Attack into white- and black-box to better present our results. The results demonstrate that the proposed MFDV-SNN maintains great performance even on the stronger attack.

**Black-box Attacks Results:** The black-box attacks include n-Pixel attack and Auto-Attack (black-box), which termed as Square attack, as reported in Table 2 (Bottom) and Table 4. For n-Pixel attack, which relies on evolutionary optimization and is derivative-free. The attack strength is controlled by the number of pixels it comprises. From the results in Table 2 (Bottom), we can see that the proposed MFDV-SNN outperforms other state-of-the-art methods in all attack strengths. Specifically, for 1−, 2−, 3− pixels attacks, compared with the best defense method Adv-BNN, we have improvements of about 24.5%, 24.5%, and 27.3%, respectively. Furtherly, we add a stronger 5-pixel attack to evaluate the proposed MFDV-SNN's efficiency. Moreover, Auto-Attack (black-box) results also confirm the strong adversarial robustness of MFDV-SNN against black-box attacks.

Overall, various white- and black-box attacks show that the proposed MFDV-SNN is strong enough. It is worth emphasizing that the proposed MFDV-SNN does not need adversarial training, while the other state-of-the-art defense methods all need adversarial

**Table 5: Comparison of the proposed MFDV-SNN with state-of-the-art defense methods. All competitors evaluate their models on the untargeted PGD attack on CIFAR-10. AT: Use of adversarial training. Results show that our proposed MFDV-SNN does not need adversarial training and achieves state-of-the-art results on clean and robust accuracy among defense algorithms.**

| Methods | Backbone | AT | Clean | PGD |
|---|---|---|---|---|
| PGD AT [23] | ResNet-20 (4x) | ✓ | 87.0 | 46.1 |
| RSE [21] | ResNext | ✗ | 87.5 | 40.0 |
| DP [20] | 28-10 Wide ResNet | ✗ | 87.0 | 25.0 |
| TRADES [34] | ResNet-18 | ✓ | 84.9 | 56.6 |
| PCL [24] | ResNet-110 | ✓ | 91.9 | 46.7 |
| PNI [12] | ResNet-20 (4x) | ✓ | 87.7 | 49.1 |
| Adv-BNN [22] | VGG-16 | ✓ | 77.2 | 54.6 |
| L2P [16] | ResNet-18 | ✓ | 85.3 | 56.3 |
| MART [30] | ResNet-18 | ✓ | 83.0 | 55.5 |
| BPFC [1] | ResNet-18 | ✗ | 82.4 | 41.7 |
| RLFLAT [26] | 32-10 Wide ResNet | ✓ | 82.7 | 58.7 |
| SADS [28] | 28-10 Wide ResNet | ✓ | 82.0 | 45.6 |
| WCA-Net [9] | ResNet-18 | ✗ | 93.2 | 71.4 |
| CAS [4] | ResNet-18 | ✓ | 86.8 | 48.9 |
| SEAT [29] | ResNet-18 | ✓ | 83.7 | 56.0 |
| FSR [17] | ResNet-18 | ✓ | 83.3 | 54.8 |
| SPAT [14] | ResNet-18 | ✓ | 84.1 | 58.3 |
| RobustResNet [15] | 28-10 Wide ResNet | ✓ | 85.5 | 58.7 |
| RPF [8] | ResNet-18 | ✓ | 83.8 | 61.3 |
| **MFDV-SNN** | ResNet-18 | ✗ | **93.7** | **79.6** |

training. The experiment results also demonstrate that the proposed MFDV-SNN does not sacrifice clean data accuracy.

## 4.4 Comparison with State-of-the-art Defenses

We compare MFDV-SNN with state-of-the-art defense methods proposed in recent years. Among these defense methods, some methods are SNNs, and some are not. We present the results in Table 5. All experiments are conducted on CIFAR-10 with the PGD attack. AT means using adversarial training. The results show that many defense methods need adversarial training, which requires a high computational cost. The proposed MFDV-SNN outperforms defense methods listed in Table 5, even with deeper network architectures, and achieves the highest accuracy on clean data and strong adversarial robustness.

## 5 ADDITIONAL ANALYSIS

### 5.1 Inspection of Gradient Obfuscation

Athalye et al. [2] claimed that some stochastic algorithms are false defense methods. These methods mainly obfuscate the gradient information to improve the model's robustness, and they propose a checklist to identify whether a defense is of an obfuscation gradient. Once the defense belongs to gradient obfuscation, it can be attacked by the proposed EOT attack [3]. In this section, we thoroughly inspect whether the proposed method belongs to gradient

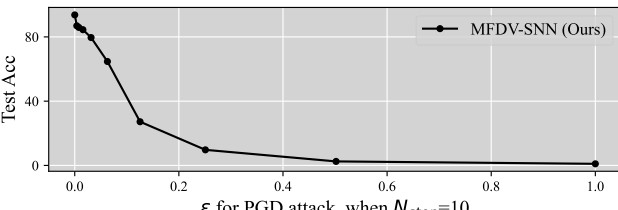

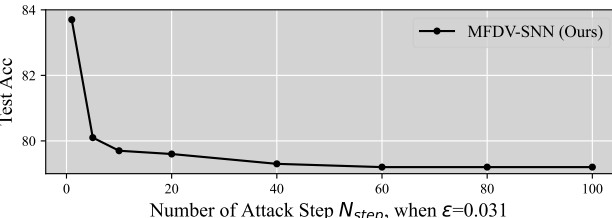

**Figure 3: On the CIFAR-10 test set, the perturbed-data accuracy of ResNet-18 under PGD attack (Upper) versus attack bound $\epsilon$, and (Bottom) versus the number of attack steps $N_{step}$.**

obfuscation by the two-stage inspection. In the first stage, we conduct experiments to check the gradient obfuscation list proposed by Athalye [2], and experiment details follow [16]. In the second stage, we directly evaluate our method by the EOT attack.

**First Stage Inspection: Checklist**

*Criterion 1:* One-step attack performs better than iterative attacks.
*Refutation:* The PGD attack is an iterative attack, and FGSM is a one-step attack. From Table 1, we can see that the accuracy of MFDV-SNN against FGSM attack is consistently higher than that of PGD attack.

*Criterion 2:* Black-box attacks are better than white-box attacks.
*Refutation:* With the development of adversarial attack methods, there are strong white-box attacks and black-box attacks, making it difficult to have a common result. Nevertheless, we can partially confirm in Table 4 that stronger black-box attacks perform worse than strong white-box attacks.

*Criterion 3:* Unbounded attacks do not reach 100% success.
*Refutation:* Following [12], as drawn in Figure 3, we run experiment by increasing the distortion bound-$\epsilon$. The results show that the unbounded attack leads to 0% accuracy.

*Criterion 4:* Random sampling finds adversarial examples.
*Refutation:* Following [12], the prerequisite is that the gradient-based (e.g., PGD and FGSM) attack cannot find adversarial examples. However, Figure 3 indicates that when we increase the distortion bound, our method can still be broken.

*Criterion 5:* Increasing the distortion bound does not increase the success rate.
*Refutation:* The experiment in Figure 3 shows that increasing the distortion bound improves the attack success rate.

**Second Stage Inspection: EOT-Attack**

Following [16, 25], we use a Monto Carlo method which expects the gradient over 80 simulations of different transformations on the CIFAR-10 dataset with the backbone ResNet-18. Experiment

**Table 6: Generalization study on different sizes of datasets. We evaluate MFDV-SNN on six datasets: MNIST, SVHN, CIFAR-10, CIFAR-100, Tiny-ImageNet and Imagenette. We use ResNet-18 as a backbone (Except for the MNIST dataset). Results show that the proposed method generalizes well on different sizes of datasets.**

| Model | MNIST | | | SVHN | | | CIFAR-10 | | |
|---|---|---|---|---|---|---|---|---|---|
| | Clean | FGSM | PGD | Clean | FGSM | PGD | Clean | FGSM | PGD |
| No Defense | 99.3 | 33.5 | 18.1 | 94.9 | 18.6 | 5.9 | 92.9 | 21.3 | 2.3 |
| MFDV-SNN | 99.2 | **93.4** | **64.1** | 94.0 | **86.1** | **82.7** | 93.7 | **85.7** | **79.6** |
| Model | CIFAR-100 | | | Tiny-ImageNet | | | Imagenette | | |
| | Clean | FGSM | PGD | Clean | FGSM | PGD | Clean | FGSM | PGD |
| No Defense | 68.8 | 12.8 | 1.5 | 48.0 | 9.0 | 1.0 | 75.5 | 9.5 | 0.0 |
| MFDV-SNN | 69.4 | **47.1** | **37.3** | 48.1 | **26.0** | **19.1** | 75.5 | **34.5** | **24.2** |

**Table 7: Generalization study on different network architectures. We evaluate MFDV-SNN on various network architectures. Results demonstrate that MFDV-SNN generalizes well on various mainstream architectures.**

| Model | VGG19 | | GoogLeNet | | DenseNet121 | |
|---|---|---|---|---|---|---|
| | FGSM | PGD | FGSM | PGD | FGSM | PGD |
| No Defense | 16.2 | 3.7 | 14.6 | 1.4 | 13.9 | 0.2 |
| MFDV-SNN | **64.1** | **53.2** | **80.2** | **66.8** | **80.1** | **71.3** |
| Model | ResNet-32 | | ResNet-44 | | ResNet-56 | |
| | FGSM | PGD | FGSM | PGD | FGSM | PGD |
| No Defense | 11.4 | 0.0 | 19.5 | 1.3 | 19.7 | 1.2 |
| MFDV-SNN | **62.7** | **57.4** | **66.6** | **64.8** | **75.0** | **71.6** |
| Model | ResNet-20[1.5×] | | ResNet-20[2×] | | ResNet-20[4×] | |
| | FGSM | PGD | FGSM | PGD | FGSM | PGD |
| No Defense | 8.1 | 0.0 | 7.3 | 0.0 | 10.7 | 0.3 |
| MFDV-SNN | **70.1** | **66.9** | **77.7** | **73.6** | **83.9** | **77.0** |

results show that PNI, Adv-BNN, and L2P can provide 48.7%, 51.2%, and 53.3% robustness, respectively. The proposed MFDV-SNN can have 79.2% robustness, which is higher than these non-gradient obfuscation methods.

Moreover, we adopt 15 MC sampling expectations of the Gaussian noise in the test phase, verified in the experiments to be stable enough to ensure the experiment results are not stochastic gradient. Overall, the proposed method passes the two-stage inspection of gradient obfuscation, ensuring that the proposed MFDV-SNN is not of gradient obfuscation.

## 5.2 Inspection of Generalization

We conduct experiments on different datasets and network architectures to evaluate the proposed MFDV-SNN generalization ability. The results are reported in Table 6 and Table 7. In Table 6, we mainly explore the datasets' influence on the proposed MFDV-SNN. Six different datasets are used in this experiment. The experiments are based on the backbone ResNet-18, except for the MNIST dataset.

We evaluate MNIST dataset performance on the LeNet architecture. The result shows that MFDV-SNN has a great generalization to different datasets. In Table 7, we mainly explore the impact of the network architectures on the proposed MFDV-SNN. Specifically, we adopt various architectures, including VGG, GoogLeNet, DenseNet, and ResNets. The results demonstrate that MFDV-SNN generalizes well on various network architectures. We also confirm that MFDV-SNN generalizes well on different network widths and depths in Table 1.

Overall, we explore the generalization of the proposed MFDV-SNN from two perspectives: network architectures and datasets. For network architectures, we further explore the effect of different types of networks and the effect of different widths and depths. Both results show the proposed MFDV-SNN generalizes well to the mainstream DNN models.

## 5.3 Comparison of Computation Time

We compare the computation time among the proposed MFDV-SNN, No defense model, and Adversarial training model. Three networks and three datasets are used. The networks include ResNet-18, ResNet-50, and WRN-34-10. The datasets include SVHN, CIFAR-10, and CIFAR-100. The results are reported in Table 8. Specifically, we report the average epoch time consumption as the evaluation metric. We have the following observations: 1) Our proposed MFDV-SNN is computationally inexpensive and close to standard defense-free training. We can observe this phenomenon in all the comparison experiments. 2) Compared to AT, we save computation time from 6.5 to 7.5 times, which makes sense since we do not need to generate adversarial examples. 3) Although different architectures are compared, the basic rules are almost identical. As the network structure becomes more complex, the computation time increases proportionally. i.e., From ResNet-18 to ResNet-50 to WRN-34-10, the complexity of the model increases sequentially, and the WB-SNN computation time increases accordingly. However, the ratio of adversarial training to our proposed WB-SNN method varies relatively steadily.

## 5.4 Further Analysis

**Effectiveness of regularization loss.** We conduct experiments to evaluate the effectiveness of the proposed regularization module on the FGSM attack and PGD attack trained on the CIFAR-10

**Table 8: Comparison of computation time among the proposed MFDV-SNN, No defense , and Adversarial training. Results show that the proposed MFDV-SNN time cost is close to standard training, saving up to 7.5 × time cost than adversarial training.**

| Time (s) / epoch | ResNet-18 | | | ResNet-50 | | | WRN-34-10 | | |
|---|---|---|---|---|---|---|---|---|---|
| | SVHN | CIFAR-10 | CIFAR-100 | SVHN | CIFAR-10 | CIFAR-100 | SVHN | CIFAR-10 | CIFAR-100 |
| No Defense | ∼ 102 | ∼ 75 | ∼ 74 | ∼318 | ∼223 | ∼222 | ∼848 | ∼560 | ∼561 |
| AT | ∼ 728 | ∼ 500 | ∼500 | ∼2300 | ∼1580 | ∼1584 | ∼6137 | ∼4217 | ∼4213 |
| MFDV-SNN | ∼ 104 | ∼ 75 | ∼ 77 | ∼320 | ∼225 | ∼225 | ∼845 | ∼561 | ∼561 |
| AT / MFDV | **7.0** | **6.7** | **6.5** | **7.2** | **7.0** | **7.0** | **7.3** | **7.5** | **7.5** |

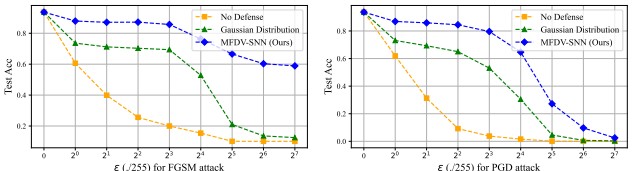

**Figure 4: Ablation study on CIFAR-10 dataset with the backbone ResNet-18. Results show the effectiveness of the proposed regularization module. FGSM attack (Left). PGD attack (Right).**

**Table 9: Ablation study on where to apply stochastic layer. We explore the effect of the number and location of stochastic layers and regularization losses on the results.**

| Model | Robust Acc |
|---|---|
| SimpleNet | 33.5 |
| + Stochastic (linear1) | 93.4 |
| + Stochastic (linear2) | 65.6 |
| + Stochastic (linear3) | 73.9 |
| + Stochastic (linear123) | **95.2** |
| + Stochastic (linear12) | 94.0 |
| + Stochastic (linear13) | 94.2 |
| + Stochastic (linear23) | 73.1 |
| ResNet-20 [4x] | 0.3 |
| + Stochastic (linear1) | 77.0 |
| + Stochastic (linear12(1)) | **78.5** |
| + Stochastic (linear12(12)) | 54.4 |

dataset with the backbone ResNet-18. The result is shown in Figure 4. No defense means that we do not add randomness to the model. Gaussian distribution means implementing the feature layer to Gaussian distribution without regularization. In practice, we achieve this by setting $\lambda_1$ to zero. MFDV indicates that we add the proposed regularization loss to the Gaussian distribution. We have the following observations. 1) Injecting stochastic into the model can improve the model robustness; for instance, Gaussian distribution has higher adversarial robustness than No defense. 2) The proposed loss function is effective; it shows that MFDV outperforms the Gaussian distribution by a large margin, while the only difference in this experiment is the proposed regularization loss.

**Where to apply stochastic layer?** We build a shallow and a deep model to explore the location to apply stochastic layers. The shallow model consists of two Conv and three linear layers, in which we implement the stochastic layer to different layers. The model robustness is evaluated on the FGSM attack. For notation, the linear1-3 denoted different layers before the last classifier layer. For example, linear1 means we implement the penultimate layer as a stochastic layer, linear2 means we implement the third layer from the classifier layer as a stochastic layer, linear12 means we implement both two layers as stochastic layers, and so forth. The result is shown in Table 9. Furtherly, we evaluate on a deeper model ResNet-20 [4x] under PGD attack, in which linear12(1) means we perform regularization at the penultimate layer, and linear12(12) means we perform regularization on both layers.

We have the following observations. 1) Applying stochastic layer to multiple high-level layers generally achieves a better performance, for instance, in shallow model, linear12 and linear13 are better than linear1, and linear123 achieves the best results. 2) Applying stochastic layer to the penultimate layer is the key to achieving strong adversarial robustness, for instance, in shallow model, linear2, linear3 and linear23 are worse than linear1, but linear12, linear13, and linear123 perform well. This makes sense because linear1 is the closest to model prediction and tends to capture label-related information. 3) Regularization on multi-layers is not suitable for large network, i.e., in ResNet-20[4x] model, linear12(1) achieves the highest robust accuracy, linear12(12) is worse than linear12(1) and linear1.

## 6 CONCLUSION

In this paper, we have theoretically demonstrated that by gradually increasing the level of stochastic noise within a deep neural network, the model inherently enhances its resistance to input perturbations. This principle guided the development of our MFDV-SNN framework, which efficiently integrates noise intensity into the model training process. Our comprehensive experiments validate the superior adversarial robustness of MFDV-SNN against established white-box and black-box attacks. Notably, the MFDV-SNN framework achieves this robustness without depending on adversarial training methods and maintains high accuracy on clean data inputs. Furthermore, the computational overhead introduced by our method is minimal, closely aligning with the costs of standard model training which underscores its practicality and effectiveness.

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
