# OpenReview forum: "Maximizing Feature Distribution Variance for Robust Neural Networks"
_acmmm.org/ACMMM/2024/Conference — MM2024 Poster_

### Official Review · Reviewer_1ia6 · 2024-05-24

**Rating:** 2
**Confidence:** 4

**Summary:**

This paper proposes a method to improve model adversarial robustness, by introducing random Gaussian noise into extracted features during training and gradually increasing the noise magnitude, which controls the uncertainty injected into the training process. The paper has evaluated the proposed method with different deep neural network architectures, datasets, and several representative adversarial attacking methods.

**Strengths:**

1.The proposed method is relatively easy to implement while providing desirable adversarial robustness, as shown in the experiments.

2.The proposed method has been evaluated from multiple perspectives, crossing datasets and models with different scales, and adversarial attacks with different attacking strengths.

**Limitations:**

1. The proposed method is not well presented. In Figure 2, it appears that the noise is injected before reaching to part for computing logits, while the analysis in section 3.2 implies that the noise injection can be applied to an arbitrary layer. Moreover, the analysis of noise injection as a regularization approach is not well presented. First of all, the underlying assumption for applying the translation operator is not specified, implying that the analysis can be applied to different layers, including layers with linear activation and various non-linear activation.

2. While there are many diverse evaluation perspectives considered in this paper, the evaluation, however, is not comprehensive enough to support the claimed performance improvement. Specifically,
(1) although one of the major contributions is that the proposed method does not require adversarial training, the experiment results on comparing the computation time required for adopting the proposed method and adversarial training do not sufficiently convince the reviewer about this claim. Specifically, to adopt the proposed method, a model needs to be trained from the very beginning with all data (the same as training a model with no defense method adopted), while for adversarial training, a previously stored checkpoint can be adopted and trained only with augmented adversarial examples. Also, it is not clear how is the computation time calculated, what is the detailed adversarial training process? How many adversarial examples are used for training? Are all models trained to obtain similar performance, or they may converge with different performances?

(2) While FGSM, PGD, C&W, and EOT attacks are mentioned in this paper, only the first two are mostly evaluated, C&W is occasionally evaluated, and EOT is only used for inspecting if the proposed method obfuscates gradients.

(3) The selection of models for evaluation appears to be quite arbitrary, for example, for evaluating computation time, a WRN-34-10 model is used yet has never been mentioned before, while previously evaluated googlenet, vgg19, and other models are ignored.

(4) As shown in Tables 2 and 4, the proposed MFDV method performs better than No Defense on clean data, it would be better if more discussion or analysis for these results could be provided, as this phenomenon has not been observed for other stochastic methods.

(5) The evaluation results shown in Table 5 cannot well justify that the proposed method outperforms SOTA defense methods, as the models adopted for comparison do not even have the same architecture.

**Suitability:**

3

---

### Official Review · Reviewer_WrBz · 2024-05-25

**Rating:** 5
**Confidence:** 4

**Summary:**

This paper theoretically shows that by gradually increasing the level of stochastic noise in a DNN, the model naturally becomes
more resistant to input perturbations. The proposed method does not require adversarial training and does not sacrifice clean data accuracy.

**Strengths:**

1. Impressively, I have carefully checked the theoretical derivation and it's indeed convinced me from a theoretical standpoint.
2. The method surpasses the performance of current state-of-the-art (SOTA) methods, showcasing its potential.
2. The proposed method is underpinned by rigorous mathematical proofs, adding to its credibility.
3. The paper is commendably articulated and easy to follow.
4. This work commits the source code and increases its repeatability.

**Limitations:**

1. The theoretical derivation of this paper is confusing with some symbols, and the meaning of some symbols is unclear, i.e. $𝑊_{SN}$, $\epsilon𝑊_{SN}$, $ 𝑊_{SN} \epsilon$. I recommend that the author clarify the theoretical section; nevertheless, this does not detract from my view that it represents a significant and valuable contribution.
2. For what I want to point out specifically,  I suggest the author add some background on **translation operator** as it is a critical point in the derivation and may take a lot of extra time to understand the principles behind this if the reader does not have a relevant background in advanced linear algebra.
2. Ablation studies on $ln(\sigma)$ and $L_{2}$ regularization penalty term in Eq.11 should be further conducted as  $L_{2}$ regularization can also enhance the robustness of the model in principle. This can help us understand the true robustness gained from **stochastic noise (SN)** introduced in the training phase.

Overall, after carefully checking the theoretical derivation of this paper, I believe that this paper demonstrates its superiority both theoretically and experimentally. It is a work with a clear motivation, and coherent theory, and achieves new SOTA performance.

**Suitability:**

3

---

### Official Review · Reviewer_nA3i · 2024-05-26

**Rating:** 4
**Confidence:** 2

**Summary:**

The paper proposes a novel method, MFDV-SNN, that improves the adversarial robustness of DNNs by gradually increasing the stochastic noise during training. This method does not require adversarial training, thus preserving the model's accuracy on clean data and significantly reducing computation time. The effectiveness of MFDV-SNN is validated through extensive experiments against various adversarial attacks.

**Strengths:**

The introduction of stochastic noise to maximize feature distribution variance is a novel method that enhances adversarial robustness.
It provides theoretical insights into how the stochastic noise injection enhances robustness and avoids the computationally expensive adversarial training, making it more efficient while maintaining clean data accuracy. Comprehensive experiments on multiple datasets and various attacks demonstrate the effectiveness of MFDV-SNN. Gradient obfucation is also inspected.

**Limitations:**

1. **Insufficient Evaluation**: The evaluation using AutoAttack is only compared to the no-defense baseline. The improvement over other baseline defenses is difficult to assess, as PGD and FGSM are relatively weaker attacks and both suffer from the gradient vanishing issue.
2. **Outdated Baseline**: To demonstrate superiority, the baseline should be updated. For instance, the selected adversarial training method is outdated and simplistic. Many recent adversarial training methods focus on improving training efficiency[a,b].
[a] Adversarial Training for Free!
[b] Fast is Better than Free: Revisiting Adversarial Training.

**Suitability:**

2

---

### Meta-Review · Area_Chair_dedd · 2024-07-01

**Recommendation:** Accept (Poster)
**Confidence:** 5

**Metareview:**

This work is appreciated by the reviewers in terms of the technical novelty and the performance improvement comparing with the state-of-the-art methods. Some concerns are related to a theoretical proof of the proposed method, which requires a more detailed explanation to be provided in the camera ready. Thus, I recommend the acceptance as a poster work.